# Error Uncertainty Analysis in Planar Closed-Loop Structure with Joint Clearances

**Yushu Yu** [1,*], **Jinglin Li** [2], **Xin Li** [3] **and Yi Yang** [4]

1    School of Mechatronical Engineering, Beijing Institute of Technology, Beijing 100081, China
2    CASIC Space Engineering Development Co., Ltd., Beijing 100039, China; jinglinli31@gmail.com
3    School of Mechanical Engineering and Automation, Beihang University, Beijing 100191, China;
     liford@163.com
4    School of Mechatronic Engineering and Automation, Shanghai University, Shanghai 200444, China;
     yiyangshu@t.shu.edu.cn
*    Correspondence: yushu.yu@bit.edu.cn

**Abstract:** For planar closed-loop structures with clearances, the angular and positional error uncertainties are studied. By using the vector translation method and geometric method, the boundaries of the errors are analyzed. The joint clearance is considered as being distributed uniformly in a circle area. A virtual link projection method is proposed to deal with the clearance affected length error probability density function (PDF) for open-loop links. The error relationship between open loop and closed loop is established. The open-loop length PDF and the closed-loop angular error PDF both approach being Gaussian distribution if there are many clearances. The angular propagation error of multi-loop structures is also investigated by using convolution. The positional errors of single and multiple loops are both discussed as joint distribution functions. Monte Carlo simulations are conducted to verify the proposed methods.

**Keywords:** closed-loop structure; joint clearance; angular and positional error; probability density function

## 1. Introduction

Rigid links are often hinged together as a fixed structure to fulfill the requirements of various tasks. If the structure is transformed from a deployable mechanism, which is widely used in the space environment, such as the solar panel array system or an antenna support structure, the possible joint clearances may have a great effect on its accuracy.

Open-loop or serial mechanisms with joint clearances have been studied recently. Pandey and Zhang presented a method for computing the positional reliability of a manipulator with random joint clearances [1]. The joint axis orientation and link length uncertainties were studied for a robot tolerance analysis and multibody dynamics method was used [2]. Based on advanced first order second moment method, the position and orientation kinematical reliabilities have been analyzed for a RRR manipulator [3]. Jawale and Thorat [4–6] have studied the maximum positional errors of open chain and closed chain manipulators in their researches. The kinematic accuracy reliability problem was also analyzed by using hybrid dimension reduction method [7]. The mechanism kinematics and dynamics uncertainties caused by a number of errors have been studied by Rao [8] and Lai [9], respectively. Flores and Lankarani [10] have established the dynamic behavior model of multibody systems with clearance joints. The joints in frictionless, lubricated and dry situations were discussed respectively [11–15]. Furthermore, the dynamics of flexible-rigid multibody systems [16,17] and flexible mechanisms [18,19] with joint clearances has also been studied. Tsai and Lai used screw method to calculate the kinematics error of multi-loop linkages [20,21]. Similarly, the maximum parallel platform error was also investigated by using the screw method [22]. Furthermore, the inertial force and

external force were considered [23,24]. In most of these studies, statics or dynamics are used to calculate the angular or positional errors, which obviously is a kinematics problem. Venanzi and Parenti-Castelli have proposed a method to determine the maximum displacement without considering force or torque [25]. Ting, Zhu and Watkins have shown that the same dimension joint clearances contributed equally to the direction error in a single loop structure [26]. Wu and Rao used interval method to analyze the tolerances and clearances [27]. Other methods include Lie group and Lie algebra method, direct linearization method [28–31], etc. In the authors' previous work, the uncertainty angular error of multiple-loop structures especially for an extendible support structure (ESS) is analyzed [32]. But the pin (journal) of the clearance joint is assumed only goes around the hole (bearing) inner surface.

In a mechanism system with uncertain joint clearances, besides the maximum error, we are also interested in the error probabilities. The uncertainty analysis for closed-loop, especially the multi-loop structure, is more difficult. Compared to the previous research, a more general error analysis methodology for closed-loop structures is proposed in this paper. Firstly, the clearance model is more general. The pin is assumed can stay anywhere inside the hole uniformly. Secondly, the positional uncertainty error analysis is added. By using the proposed virtual link projection method, the angular error and the positional error of single-loop structures are determined with their probability density functions (PDF), and the angular error propagation of multi-loop structures is also studied. Finally, the multi-loop positional error is investigated. In order to verify the proposed theoretical methodology, Monte Carlo simulations are conducted and compared to the theoretical results.

The obtained results can be used to estimate how much the joint clearance affects the error of the planar closed-loop structure. We will prove that the clearance of each joint in the closed loop has the same effect on the overall error theoretically. This provides theoretical basis for the manufacture of such planar closed-loop structure.

This paper is organized as follows. The general closed-loop error model is analyzed in Section 2. The error PDF analysis and the numerical simulation method is presented in Section 3. The comparison results of the numerical simulation and the theoretical results are shown in Section 4.

## 2. Closed-Loop Error Model Analysis

### 2.1. Modelling Method

In a support structure, the stable triangle configuration is commonly applied. If a deployable mechanism is locked as such a structure, the joint clearances are often introduced inevitably. The following models are established according to this situation. We have already known that the straight-line locked joints have a negligible error effect and a multiple-joint can be simply seen as a series of single-joints [32]. Basically, a single clearance joint needs to be studied.

A clearance revolute joint can be expressed as shown in Figure 1. The shaft center can stay anywhere freely when $k < R - r$ if all components are rigid.

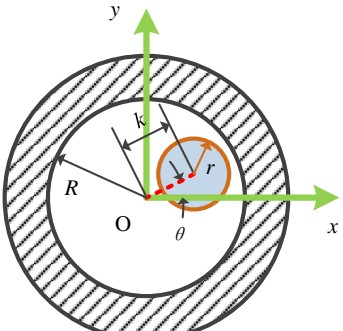

**Figure 1.** The geometrical representation of clearance joints.

In the space floating environment, it is reasonable to assume that the PDF of the pin center is a uniform distribution function, and it can be given as

$$\rho(x,y) = \begin{cases} \frac{1}{\pi K^2}, & (K = R - r, x^2 + y^2 \leq K^2) \\ 0, & \text{(otherwise)} \end{cases} \tag{1}$$

where, $(x,y)$ is the coordinate of the pin center in frame $xOy$.

For a closed-loop structure with clearance joints, we could find two common expressions of hinged joint with clearance, as shown in Figure 2.

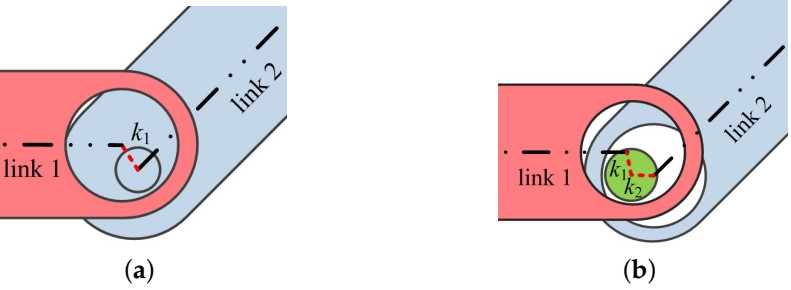

(**a**)                    (**b**)

**Figure 2.** Two expression types of hinged joint with clearance. (**a**) Type 1. (**b**) Type 2.

The pin is connected to link 2 of type 1, while the pin of type 2 is an individual part and it is widely used. However, type 2 can be seen as a connection of two sets of type 1. For this reason, a basic model of a closed-loop structure with joint clearances is established as shown in Figure 3.

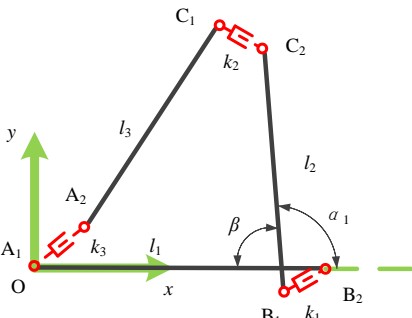

**Figure 3.** The closed-loop model with joint clearances.

In this model, each joint clearance with its adjacent links can be seen as a RPR (Revolute joint–Prismatic joint–Revolute joint) kinematic chain. In the closed loop, there are 8 movable links (including virtual links) and 9 joints, the degree of freedom (DoF) thus is calculated as [33]

$$F = 3 \times 8 - 2 \times 9 = 6 \tag{2}$$

Each clearance has two DoFs, means its direction and size can change independently. Compared with the links, the clearance size is small enough, which ensures that the motion induced by clearance is similar to that of a crank. In the system, point coordinates are determined as

$$A_1 = \begin{bmatrix} 0 & 0 \end{bmatrix}^T \tag{3}$$

$$B_2 = \begin{bmatrix} l_1 & 0 \end{bmatrix}^T \tag{4}$$

$$B_1 = B_2 + \begin{bmatrix} x_1 & y_1 \end{bmatrix}^T \tag{5}$$

Let $\alpha_1$ denote the angle between $B_1C_2$ and axis $x$, we have

$$C_2 = B_1 + l_2 \begin{bmatrix} \cos\alpha_1 & \sin\alpha_1 \end{bmatrix}^T \tag{6}$$

$$C_1 = C_2 + \begin{bmatrix} x_2 & y_2 \end{bmatrix}^T \tag{7}$$

$$A_2 = A_1 + \begin{bmatrix} x_3 & y_3 \end{bmatrix}^T \tag{8}$$

According to the structural constraint,

$$l_3 - \|A_2C_1\| = 0 \tag{9}$$

where, $x_i = k_i\cos\theta_i$, $y_i = k_i\sin\theta_i$. $\theta_i$ is the inclined angle of $k_i$ in the ground frame. $k_i$ and $l_i$ are labelled in Figure 3. Definitely, $\alpha_i$ can be solved (two solution sets) from these equations then all point locations will be determined accordingly.

Furthermore, a multi-loop structure model is established as shown in Figure 4.

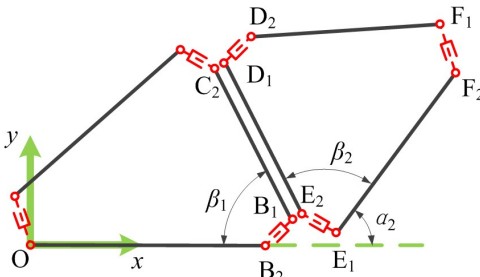

**Figure 4.** The multi-loop model with joint clearances.

Reasonable simplification is applied in the model that it can be seen as a new single-loop structure is attached to the previous model. Similar to most supporting structures, in Figure 4, links $B_1C_2$ and $E_2D_1$ overlap each other. Points $B_1$ and $C_2$ have been solved from the single-loop model, and in the new situation, they can be replaced by points $E_2$ and $D_1$, respectively. Then the loop $D_1E_2E_1F_2F_1D_2$ corresponds to the loop $A_1B_2B_1C_2C_1A_2$. Accordingly, all coordinates of the interested points are solved. It is also fine to attach more loops in a similar manner. In the next subsection, simpler methods will be discussed to determine the boundaries of the angular and positional errors. However, further statistic simulations are on the basis of the proposed models in this subsection.

### 2.2. The Angular Error and Positional Error Boundaries

By using optimization methods, the extremal angular and positional errors are obtained quickly. The boundary calculation is actually a workspace problem, and it is not easy to be determined. In this section, a direct way is presented based on vector translations. In Figure 3, we have $\beta = \psi - \alpha_1$, which is also shown in Figure 5.

All line segments in Figure 5 are seen as vectors. After the translations, $\beta$ is not changed. The maximum and minimum of $\beta$ are shown in (a) and (b) in Figure 6, respectively.

The angular error boundaries caused by joint clearances, obviously are determined by the extremal values. The equivalent opposite link to $\beta$ is a critical factor. $\beta$ grows larger if the length of the opposite link becomes longer. This conclusion is also suitable for multi-loop situations. For example, in Figure 4, $\alpha_2 = \pi - \beta_1 - \beta_2$. $\beta_1$ and $\beta_2$ are in separated single loops, and their maximal values can be obtained accordingly to minimize $\alpha_2$.

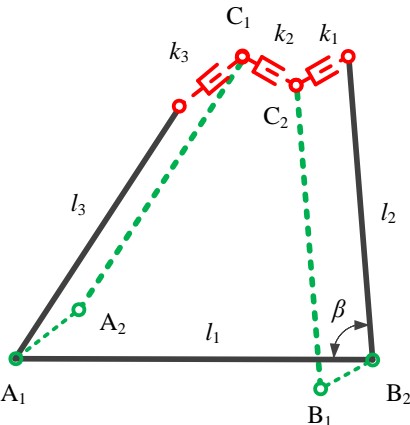

**Figure 5.** Equivalent $\beta$ after vector translations.

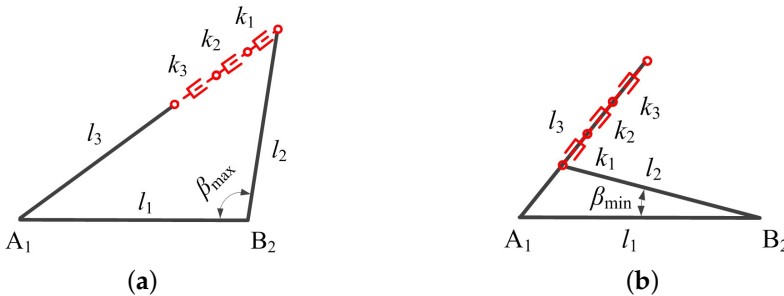

**Figure 6.** The extremal configurations. (**a**) Maximum. (**b**) Minimum.

For the positional error boundary analysis, the workspace of point $C_1$ is studied in Figure 7 first.

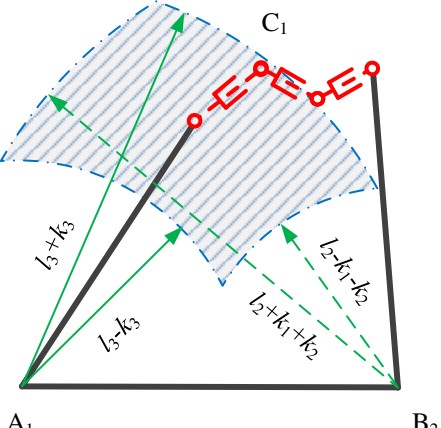

**Figure 7.** Workspace of point $C_1$.

In Figures 5 and 7, $C_1$ is the intersection point of $A_1C_1$ and $B_2C_1$. As can be seen, the length of the two line segments varies in the intervals of $[l_3 - k_3, l_3 + k_3]$, $[l_2 - k_1 - k_2, l_2 + k_1 + k_2]$. As a result, $C_1$ must lie in the contour shading area. The four intersected points are called key points here. While for multi-loop structures, the positional analysis is more complicated. Possible locations of point $F_1$ are analyzed in Figure 8.

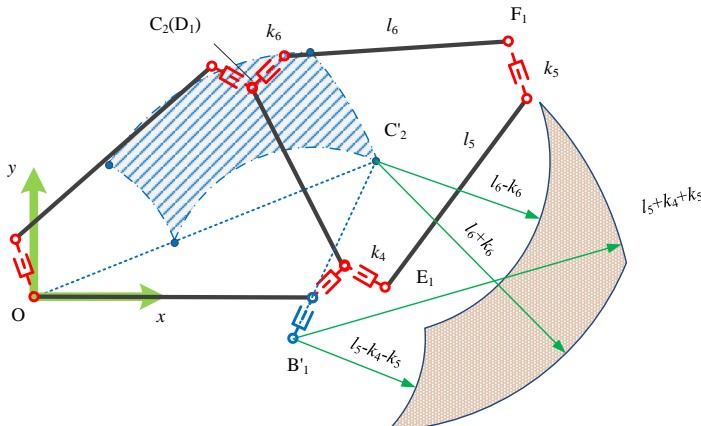

**Figure 8.** The positional analysis for multi-loop structures.

If $C_2$ stays at $C_2'$, $B_1$ will move to $B_1'$. Based on the analysis of Figure 7, an area section of $F_1$ with four new key points is drawn in Figure 8. Similarly, if $C_2$ moves to other three key points, we could find 16 new key points in total for $F_1$, as shown in Figure 9.

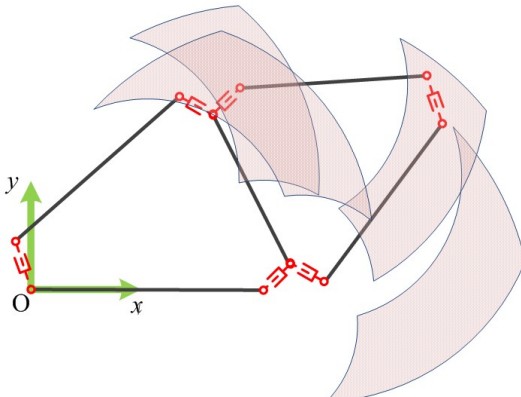

**Figure 9.** Area sections of $F_1$ with 16 key points.

The workspace of $F_1$ can be expressed approximately by an area that could encircle the 16 points. In fact, the clearance size is much smaller than the links, so all the arcs between the key points can be replaced by straight lines. Based on the above knowledge, the boundary problems are simplified accordingly.

## 3. Error PDF Analysis and Simulation Method

In the last section, we have investigated a convenient method of estimating error boundaries. Furthermore, the error probability density function (PDF) is always a challenging problem, especially for closed-loop structures. In order to further analyze the PDF, we present the equivalent opposite link to $\beta$ in dashed line as shown in Figure 10.

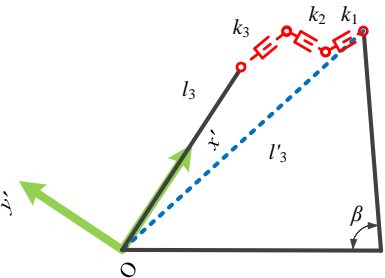

**Figure 10.** Equivalent opposite link.

The equivalent link includes many uncertain errors and it needs to be simplified. The projection method is applied for this purpose. The adjacent connected error virtual links $k_i$ are projected onto line $l_3$ respectively, in order to simplify the expressions. Figure 11 shows such simplification. We use $k_3'$ to represent the new error virtual link transformed from the three error virtual links. It is seen that the sum of the length of $l_3$ and the projection lines will replace the length of $l_3'$ as $l_3' = \sqrt{\left(l_3 + k_{3x}'\right)^2 + {k_{3y}'}^2}$, which includes ${k_{3y}'}^2$. As $k_{3y}'$ is a small item, its square is negligible. The simplified expression of $l_3'$ can thus be written as

$$l_3' = \sqrt{\left(l_3 + k_{3x}'\right)^2 + {k_{3y}'}^2} \approx l_3 + k_{3x}' \tag{10}$$

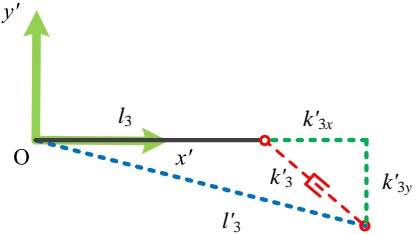

**Figure 11.** Representation of equivalent linkage length.

In a uniformly distributed circle as shown in Figure 12, the PDF along axis $x$ can be calculated as

$$
\begin{aligned}
\int_{-\infty}^{+\infty} \rho(x_1)\mathrm{d}x_1 &= \int_{-K}^{K} c \cdot 2\sqrt{K^2 - x_1^2} \cdot \mathrm{d}x_1 \\
&= c\left[x_1\sqrt{K^2 - x_1^2} + K^2 \arcsin \frac{x_1}{K}\right]\Bigg|_{-K}^{K} \\
&= c \cdot \pi K^2 = 1
\end{aligned} \tag{11}
$$

where

$$c = \frac{1}{\pi K^2} \tag{12}$$

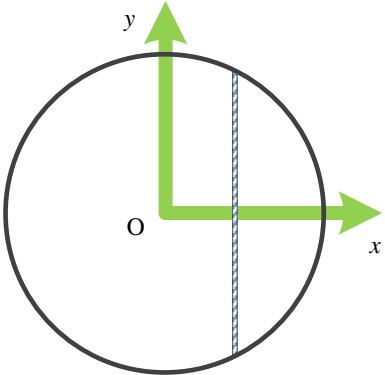

**Figure 12.** The PDF along axis $x$.

Therefore we have,

$$\rho_1(x_1) = \begin{cases} \frac{2}{\pi K^2}\sqrt{K^2 - x_1^2}, & x_1 \in [-K, K] \\ 0, & \text{otherwise} \end{cases} \tag{13}$$

where $x_1$ is the $x$ component of the clearance $-1$. Accordingly, the average and variance of $x_1$ are

$$\mu_1 = \int_{-\infty}^{+\infty} x_1 \cdot \rho_1 \cdot \mathrm{d}x_1 = \int_{-K}^{K} x_1 \cdot \frac{2}{\pi K^2}\sqrt{K^2 - x_1^2} \cdot \mathrm{d}x_1 = 0 \tag{14}$$

$$\sigma_1^2 = \int_{-\infty}^{+\infty} x_1^2 \cdot \rho_1 \cdot \mathrm{d}x_1 = \int_{-K}^{K} x_1^2 \cdot \frac{2}{\pi K^2} \sqrt{K^2 - x_1^2} \cdot \mathrm{d}x_1 = \frac{K^2}{4} \tag{15}$$

If there are more joint clearances, the propagation of them in convolution form is as

$$\rho_{1,2}(X) = \rho_1 * \rho_2(X) = \int_{-\infty}^{+\infty} \rho_1(x_1) \cdot \rho_2(X - x_1) \cdot \mathrm{d}x_1 \tag{16}$$

$$\rho_{1,2,\dots,n}(X) = \rho_1 * \rho_2 * \cdots * \rho_n(X) \tag{17}$$

where,

$$X = \sum_{i=1}^{n} x_i \tag{18}$$

where $x_i$ is the $x$ component of the clearance $-i$.

Substituting Equations (13)–(16), then the integral equation is difficult to solve even by using numerical methods. Fortunately, if we assume all joint clearances are in the same size, according to Lindeberg-Levy central limit theorem, Equation (17) can be expressed as

$$\rho_{1,2,\dots,n}(X) \approx \frac{1}{\sqrt{2n\pi}\sigma_1} e^{\frac{-X^2}{2n\sigma_1^2}} \tag{19}$$

As can be seen, the length error of a single link with many clearances will approach the Gaussian distribution. The relationship between the open link and the closed-loop needs to be bridged. If $y = f(x)$, then

$$\rho_y(Y) = \rho_x\left(f^{-1}(Y)\right) \left| \frac{\mathrm{d}f^{-1}(Y)}{\mathrm{d}Y} \right| \tag{20}$$

From Figure 10 and Equation (10), we already have

$$l_3' = \sqrt{l_1^2 + l_2^2 - 2l_1 l_2 \cos\beta} \approx l_3 + X \tag{21}$$

Then according to Equation (20), we arrive at,

$$\rho(\beta) = \rho\left(\sqrt{l_1^2 + l_2^2 - 2l_1 l_2 \cos\beta} - l_3\right) \cdot \frac{l_1 l_2 \sin\beta}{\sqrt{l_1^2 + l_2^2 - 2l_1 l_2 \cos\beta}} \tag{22}$$

This equation can be simplified. Let $\beta_0$ be the nominal value of the studied angle, and

$$\beta \approx f(X) = \arccos \frac{l_1^2 + l_2^2 - (l_3 + X)^2}{2l_1 l_2} \tag{23}$$

By using the Taylor series expansion at $X = 0$, we get

$$\beta = \arccos \frac{l_1^2 + l_2^2 - l_3^2}{2l_1 l_2} + \frac{l_3 + X}{l_1 l_2 \sqrt{1 - \left(\frac{l_1^2 + l_2^2 - (l_3 + X)^2}{2l_1 l_2}\right)^2}} X \tag{24}$$

If the small coefficients of $X$ are ignored, we can obtain

$$\varepsilon_1 = \beta - \beta_0 \approx \frac{l_3}{l_1 l_2 |\sin\beta_0|} X \tag{25}$$

where, $\epsilon_1$ denotes the angular error. As can be seen from Equation (25), $\epsilon_1$ has the same PDF form with $X$, and

$$\mu(\varepsilon_1) = \frac{l_3}{l_1 l_2 |\sin\beta_0|} \mu(X) = 0 \tag{26}$$

$$\sigma^2(\varepsilon_1) = \left(\frac{l_3}{l_1 l_2 \sin \beta_0}\right)^2 \sigma^2(X) = n\left(\frac{l_3 K}{2 l_1 l_2 \sin \beta_0}\right)^2 \tag{27}$$

For a multi-loop structure, as shown in Figure 4, a single loop is stacked on the other. Similar to Equation (16), the propagation angular error $E$ is

$$\rho_{1,2}(E) = \rho_1 * \rho_2(E) = \int_{-\infty}^{+\infty} \rho_1(\varepsilon_1) \cdot \rho_2(E - \varepsilon_1) \cdot d\varepsilon_1 \tag{28}$$

Particularly, if the two PDFs are both in Gaussian distribution, such as

$$\begin{cases} \rho_1(\varepsilon) \sim N\left(\mu_{\varepsilon1}, \sigma_{\varepsilon1}^2\right) \\ \rho_2(\varepsilon) \sim N\left(\mu_{\varepsilon2}, \sigma_{\varepsilon2}^2\right) \end{cases} \tag{29}$$

Then the convolution also will be in Gaussian distribution as

$$\rho_{1,2}(E) \sim N\left(\mu_{\varepsilon1} + \mu_{\varepsilon2}, \sigma_{\varepsilon1}^2 + \sigma_{\varepsilon2}^2\right) \tag{30}$$

It is obvious that the analysis can be extended to the structures with more loops.

Compared to the angular distribution, the positional PDF is a two-dimensional problem and is more complicated. For a single-loop, the calculation is simplified by using the proposed projection distribution method. The position of point $C_1$ is given in Figure 13.

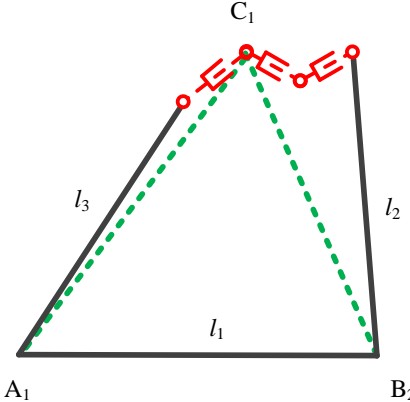

**Figure 13.** The position of $C_1$.

In Figure 13, $C_1$ is the intersection point of lines $A_1C_1$ and $B_2C_1$. The point is determined when the length of the two lines is assigned. Accordingly, the joint probability distribution for $C_1(C_{1x}, C_{1y})$ is equivalent to the joint distribution for the two independent lengths. In Figure 13,

$$l_{AC} = \sqrt{(C_{1x} - A_{1x})^2 + (C_{1y} - A_{1y})^2} \approx l_3 + X_1 \tag{31}$$

$$l_{BC} = \sqrt{(C_{1x} - B_{1x})^2 + (C_{1y} - B_{1y})^2} \approx l_2 + X_2 \tag{32}$$

where, $X_1$ and $X_2$ are the clearance projections onto line $l_3$ and $l_2$, respectively. If the length relationship is written uniformly as $l' \approx l + X$, by using Equation (20), we obtain

$$\rho(l') \approx \rho(l + X) = \rho(X) \tag{33}$$

The joint distribution can be expressed simply as

$$\rho(C_{1x}, C_{1y}) = \rho(l_{AC}) \cdot \rho(l_{BC}) = \rho(l_{AC} - l_3) \cdot \rho(l_{BC} - l_2) \approx \rho(X_1) \cdot \rho(X_2) \tag{34}$$

For the multi-loop positional analysis, such as point $F_1$ in Figure 4, a more complicated method needs to be developed which will be included in our future work. However, in the following section, a specific case of multi-loop positional error is studied by fitting method.

Monte Carlo method relies on repeated random sampling and it is often used in generating draws from a probability distribution. The initial input clearance PDFs should distribute uniformly in circles, such as being shown in Equation (1). If $(x_i, y_i)$ is expressed in a polar coordinate as $(k_i, \theta_i)$, then

$$\begin{cases} x_i = k_i \cos \theta_i \\ y_i = k_i \sin \theta_i \end{cases}, k_i \in [0, K_i], \theta_i \in [0, 2\pi) \tag{35}$$

Hence we have,

$$\begin{aligned} \rho(k_i, \theta_i) &= \rho(x_i, y_i) \left| \frac{\partial(x_i, y_i)}{\partial(k_i, \theta_i)} \right| \\ &= \frac{1}{\pi K_i^2} \begin{vmatrix} \frac{\partial x_i}{\partial k_i} & \frac{\partial x_i}{\partial \theta_i} \\ \frac{\partial y_i}{\partial k_i} & \frac{\partial y_i}{\partial \theta_i} \end{vmatrix} = \frac{k_i}{\pi K_i^2} \end{aligned} \tag{36}$$

Clearly, in the polar system, the distribution density is higher with $k_i$ growing. It is not convenient when running the simulations in programs. While in Cartesian coordinates, $x_i$ and $y_i$ both distribute uniformly in the interval $[-K_i, K_i]$. If $x_i + y_i > K_i$, the result will be discarded. The method is easier and it is an important reason that the model is established in a Cartesian coordinate. Furthermore, during the simulations, solution of Equations (3)–(9) may be used. The proposed calculation process is summarized in Figure 14.

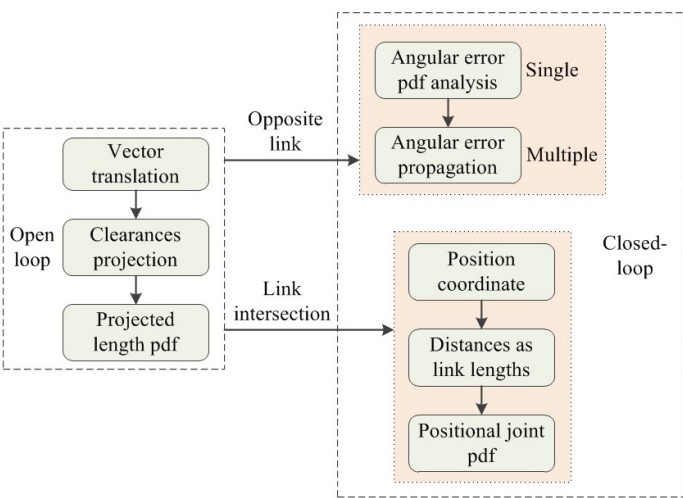

**Figure 14.** The calculation process.

## 4. Numerical Simulations

The above analysis should be verified in steps using Monte Carlo method. The joint clearance is simulated first according to the method presented at the end of last section. Let $K$ be 1 mm, then $\rho(x, y) = 1/\pi \approx 0.32$. While in Figure 15, after 105 times of randomly generating of point $(x, y)$, it is found that about $7.86 \times 104$ points are in the unit circle, and the average density of these points are about 0.31.

Along $x$ axis, if the interval $[-1, 1]$ is divided into smaller intervals, count of the point densities in the small intervals and the values are plotted as a bar chart in Figure 16. The theoretical curve according to Equation (13) is also painted in this figure.

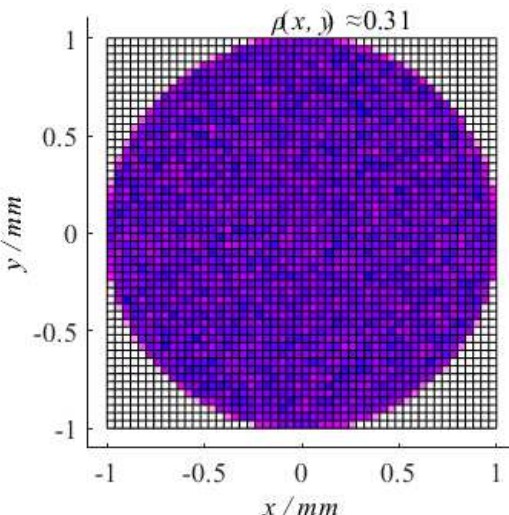

**Figure 15.** The probability density simulation of a clearance joint.

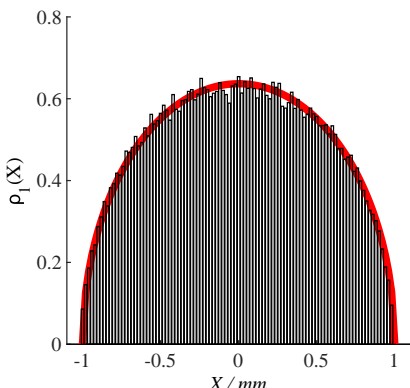

**Figure 16.** The PDF $\rho_1(X)$ along axis $x$.

If more clearances are stacked, first their densities are simulated respectively. Let $X = x_1 + x_2 + x_3$ and $X = x_1 + x_2$. Again, the statistics bar charts are plotted in Figure 17a,b.

As can be found in Figure 17, the shapes of the bars are closed to Gaussian functions. According to Equation (19), the curve of $N\,(0, 0.5)$ is plotted in Figure 17a and the curve of $N\,(0, 0.75)$ is plotted in Figure 17b. The results show that, in Equation (19), when $n \geq 2$, the equation is accurate enough.

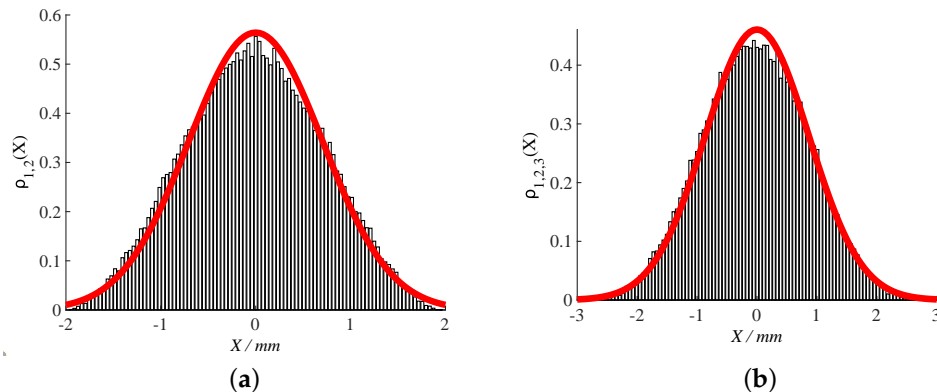

| (a) | (b) |

**Figure 17.** The PDFs along axis $x$. (a) $\rho_{1,2}(X)$ curve and bar chart. (b) $\rho_{1,2,3}(X)$ curve and bar chart.

Furthermore, the PDF of $\epsilon_1$ is tested. In this case, let $l_1$, $l_2$ and $l_3$ be 200 mm, then the nominal value $\beta_0$ is $\pi/3$. In Figure 18, the simulation result is drawn, while according to Equations (26) and (27), the Gaussian function curve $\rho_1(E)$ is obtained with $\mu(\epsilon_1) = 0$ and $\sigma_2(\epsilon_1) = 2.5 \times 10^{-5}$. In the simulations, the maximum and minimum values of $\beta$ are 1.0633 rad and 1.0314 rad, respectively. While in Figure 6, the theoretical values should be $\beta_{max} = 1.0646$ rad and $\beta_{min} = 1.0299$ rad. The results are close enough. Accordingly, the theoretical boundary of the error is determined as $[-0.0173\ \text{rad},\ 0.0174\ \text{rad}]$.

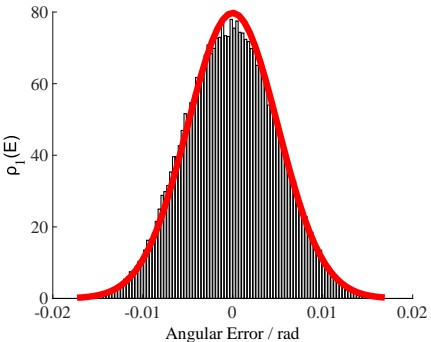

**Figure 18.** The angular error PDF of single closed-loop.

For multiple loops, assuming the two individual loops are the same, then according to Equation (30), we have $\rho_{1,2}(E)$ $(0, 5 \times 10^{-5})$, and the propagation boundary is $[-0.0346\ \text{rad},\ 0.0347\ \text{rad}]$. The function is plotted with simulation result in Figure 19.

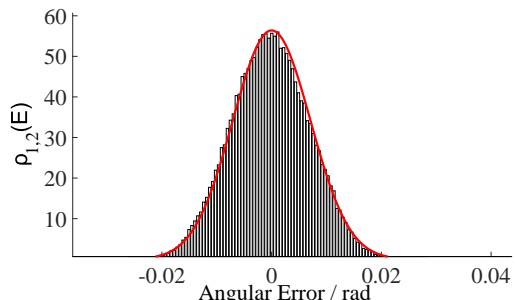

**Figure 19.** The propagation angular error PDF of two closed-loops.

The distribution statistics of point $C_1$ is as shown in Figure 20a, while by using Equation (34), the PDF of the point is calculated, as shown in Figure 20b. For a small area section, the PDF is expressed by its height.

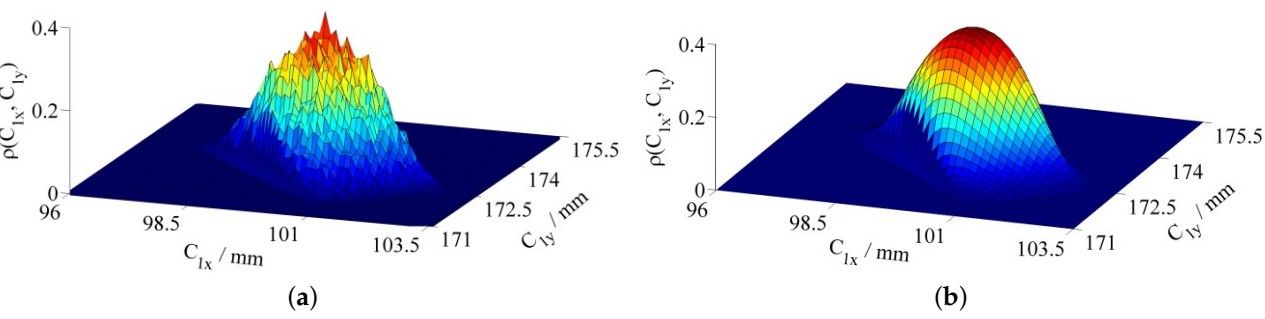

(**a**)          (**b**)

**Figure 20.** The PDF of Point $C_1$. (**a**) Simulation result. (**b**) Theoretical result.

The workspace of $C_1$ determined by the method of Figure 7 is plotted in Figure 21 with the above simulation result. The theoretical boundary is in dashed lines.

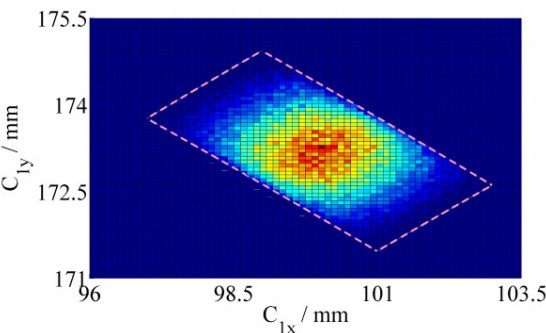

**Figure 21.** The theoretical boundary of $C_1$.

Similarly, according to Figure 9, the 16 points for the studied two loops are determined, as shown in Figure 22. The points labelled from 1 to 10 are connected in order as the simplified boundary for $F_1$. In the simulation tests for $F_1$, $7.5 \times 104$ random points are obtained and their densities are calculated. The result is also plotted in Figure 22.

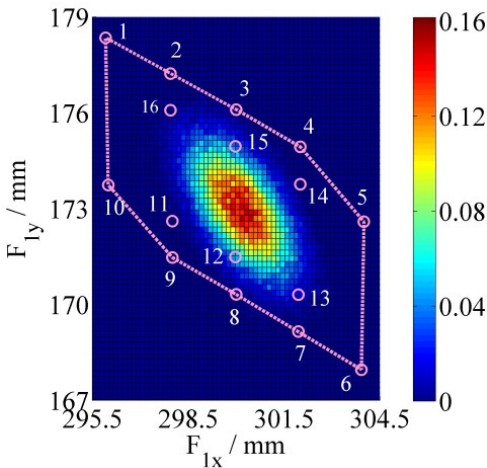

**Figure 22.** The simplified boundary and the simulation for point $F_1$.

As can be seen, nearly all the simulation points are located in the boundary, and in the boundary-neared area, the possibility of a point appearing is very small. In the previous sections, although the theoretical PDFs for multi-loop points are not given, the PDF can be obtained by the fitting method. According to the simulation data of $F_1 = (F_{1x}, F_{1y})$, the means of the coordinates (omitting measure units) are $\mu(F_{1x}) = 299.9970$ and $\mu(F_{1y}) = 173.1990$. The covariance values of the two samples are $\sigma^2(F_{1x}) = 1.0002$, $\sigma^2(F_{1y}) = 1.5003$ and $\sigma(F_{1x}, F_{1y}) = \sigma(F_{1y}, F_{1x}) = -0.7241$. In the simulation figure, the PDF can be seen as a multivariate Gaussian distribution. Accordingly, with the mean and the covariance, the fitting PDF pattern is drawn in Figure 23.

The two similar figures show that the PDF of $F_1$ is close to a two dimensional Gaussian distribution function. It is important to the future theoretical calculation work.

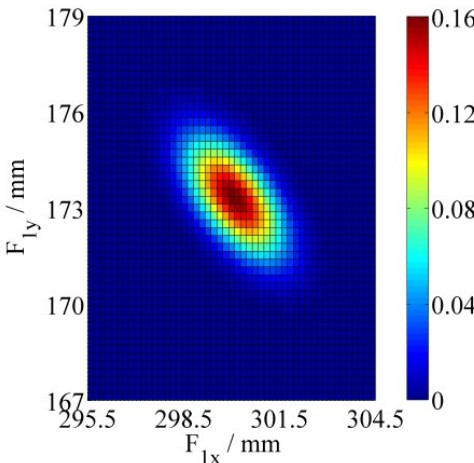

**Figure 23.** The fitting PDF pattern.

## 5. Conclusions

A general model for planar closed-loop structures with joint clearances is established to solve the uncertainty problem. By using the vector translation method, the boundaries of angular errors of both single-loop and multi-loop structures and the boundary of positional error of single-loop structures can be easily determined. Furthermore, the approximate boundary of the multi-loop positional error is studied by connecting several key points. The clearances are analyzed as uniformly distributed in a circle area. The virtual link projection method is used to calculate the PDF of link length. The length PDF approaches Gaussian distribution if there are two more random clearances. Then functions between open loop and closed-loop errors are deduced. The PDFs of the angular error and its opposite link have the same form. The positional errors are investigated as joint distributions. All of the proposed methods are verified by Monte Carlo method. Future work includes the error uncertainty analysis of such closed-loop structure under the influence of varying temperature and gravity, as the closed-loop structure is widely used in space environments.

**Author Contributions:** Conceptualization, Y.Y. and X.L.; methodology, Y.Y. (Yushu Yu) and X.L.; software, J.L.; validation, Y.Y. (Yushu Yu), J.L. and Y.Y. (Yi Yang); formal analysis, Y.Y. (Yi Yang); investigation, Y.Y. (Yushu Yu); resources, Y.Y. (Yushu Yu); data curation, J.L.; writing—original draft preparation, X.L.; writing—review and editing, Y.Y. (Yushu Yu); visualization, J.L.; supervision, Y.Y. (Yushu Yu); project administration, Y.Y. (Yushu Yu); funding acquisition, Y.Y. (Yushu Yu). All authors have read and agreed to the published version of the manuscript.

**Funding:** This work was supported by the Beijing Institute of Technology Research Fund Program for Young Scholars.

**Institutional Review Board Statement:** Not applicable.

**Informed Consent Statement:** Not applicable.

**Data Availability Statement:** Not applicable.

**Acknowledgments:** The authors would like to thank the valuable comments from the anonymous reviewers.

**Conflicts of Interest:** The authors declare no conflict of interest.

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
