# Peer review of "Error Uncertainty Analysis in Planar Closed-Loop Structure with Joint Clearances"

_metals, doi:10.3390/met11111872_

Round 1

Reviewer 1 Report

The paper presents very important research regarding the errors in joint clearances. This topic is particularly important as most of the time the mathematical models for closed-loop structures (and not only) consider the links as “perfect”.

The authors propose an error uncertainty analysis which is then applied for a planar mechanism.

The introduction is well-written covering most of the work achieved in this topic.

In equation 2 please explain the numerical values used in the computation of the mechanism DoF. For structural synthesis, you have, among others, the Gruebler formula or another formulation used by Plitea:

Plitea, N., Hesselbach, J., Pisla, D., Raatz, A., Vaida, C., Wrege, J., Burisch, A.: Innovative development of parallel robots and microrobots, Acta Tehnica Napocensis, Series of Applied Mathematics and Mechanics, vol. 5 (49), pp. 5-26, 2006

The title of figure 2 seems awkward, please rephrase that.

In figure 5 I believe that it is presented the equivalent mechanism not the equivalent “beta” which remains the same after vector translations.

What do you mean by scope of point? At least to my knowledge (apologise if I’m mistaken) I could not find the term. Of course, from the figure it is clear that you are describing the geometrical loci or let’s say “workspace” of the point C.

Please explain better the transition from the model presented in figure 10 to the one in figure 11, where it seems that the three consecutive links ki have been replaced by one. And as you discuss about error propagation is it ok to neglect the “k3y”?

The numerical analysis on the errors seems correct from my point of view.

What it is missing, in my opinion, is the discussion about the close loop mechanism. It is well known and stated in many papers and books that parallel (closed loop) mechanisms have better positioning accuracy as the joints errors do not multiply towards the end-effector. As you mention the closed loop mechanism it would be interesting to see how the errors are reduced in the mechanism as opposed to a open-loop counterpart. But maybe this is a different thing and could be approached as future research.

On the overall the paper is well-written but some parts are not clearly explained which diminishes the value of the results. I would recommend the authors to clear these non-conformities which would put the right value on the research results.

The English level is of, with very small errors throughout which with a careful read can be all eliminated. Please make sure that when you use abbreviations you write them in capital letters. (line 60 – PDF correct, paragraph 72, above the first equation – pdf is wrong).

Reviewer 2 Report

The paper deals with error uncertainty analysis in planar closed-loop structure with joint clearances. The angular and positional error uncertainties are studied in this work. A virtual link projection method is proposed to deal with the clearance affected length error probability density function for open-loop links.

Introduction includes state of the art in area of mechanisms with joint clearances with relevant links to references. Next part of the paper contains closed-loop error model analysis. Furthermore, a multi-loop structure model is established. Next part is focused to numerical simulations of solved problem.

Technically and stylistically, the article is written correctly. Selected methods of solution are suitable, and the obtained results bring new knowledge in the solved area. The graphic design is satisfactory, and the presentation of the results is clear.

Comments for improvements:

Some images (fig. 1, 3, 5, 7, 8, 10, 11, 12, 13, 14, 21, 22, 23) have enormous text compared to other images and article text. It would be good to fix it in a unified style.

The scientific article should conclude with further plans for future development. There is also a lack of discussion on the practical implementation and usability of the obtained results in practice.

I recommend to revise this paper before publication.

Round 2

Reviewer 1 Report

Based on the authors response and the modifications within the paper, I consider that this contribution can be published. 

I congratulate the authors for their work.